# REVISITING THE OTHELLO WORLD MODEL HYPOTHESIS

## ABSTRACT

Li et al. (2023) used the Othello board game as a test case for the ability of GPT-2 to induce world models, and were followed up by Nanda et al. (2023b). We briefly discuss the original experiments, expanding them to include more language models with more comprehensive probing. Specifically, we analyze sequences of Othello board states and train the model to predict the next move based on previous moves. We evaluate seven language models (GPT-2, T5, Bart, Flan-T5, Mistral, LLaMA-2, and Qwen2.5) on the Othello task and conclude that these models not only learn to play Othello, but also induce the Othello board layout. We find that all models achieve up to 99% accuracy in *unsupervised* grounding and exhibit high similarity in the board features they learned. This provides considerably stronger evidence for the Othello World Model Hypothesis than previous works.

## 1 INTRODUCTION

Li et al. (2023) used the Othello board game to probe the ability of LLMs to induce world models. Their network had a 60-word input vocabulary, corresponding to the 64 tiles of an Othello board, except for the four that are already filled at the start. They trained the network on two datasets: one on about 140,000 real Othello games and another on millions of synthetic games. They then trained 64 independent non-linear probes (two-layer MLP classifiers) to classify each of the 64 tiles into three states: black, blank, and white, using internal representations from Othello-GPT as input. The error rates of these non-linear probes dropped from 26.2% on a randomly-initialized model to only 1.7% on a trained model, while linear probes performed close to random. Li et al. (2023) saw this as evidence that LLMs can induce (non-linear) world models, at least for Othello board games, supporting the Othello World Model Hypothesis – – the hypothesis that LLMs trained on Othello move sequences can induce a relevant world model, including the Othello board layout.

Nanda et al. (2023b) did a follow-up study in which they found that linear probes also work if trained slightly differently. Instead of focusing on tile color, they probed the board state relative to the current player at each timestep, using labels such as MINE, YOURS, and EMPTY. This reduced the error rate of the probes to less than 10%. They speculated that world knowledge is often linearly represented in language models, since 'matrix multiplication can easily extract a different subset of linear features for each neuron.'

Now, training a probe as a research methodology comes with several weaknesses, including: a) probing classifiers can be prone to spurious correlations (Barrett et al., 2019). b) They do not tell us how information is arranged globally in LLMs.[1] c) They therefore only detect a subset of the interesting properties of world models, e.g., excluding the spatial relations that would enable analogical reasoning (Mikolov et al., 2013).

**Contributions** We therefore revisit the Othello World Model Hypothesis, reevaluating it using a methodology that does not suffer from weaknesses a)–c) (see Figure 1), in order to reassess the ability of LLMs to induce world models. If our results are positive, they will significantly strengthen the case for the hypothesis that LLMs induce

---

[1]Li et al. (2023) tried to compensate for this by using PCA to plot the probing classifiers in three dimensions. The PCA plots suggest that the induced global structure is meaningful, but the probing paradigm cannot quantify its meaningfulness.

world models; if not, they will suggest that the evidence cited in Li et al. (2023) and Nanda et al. (2023b) was perhaps a (spurious) effect of the probing paradigm itself.

We begin by re-modeling Othello using a range of model sizes (GPT-2, BART, T5, Flan-T5, LLaMA-2, Mistral, Qwen2.5), as prior research has predominantly focused on smaller models like GPT-2. We retrain these models using game data of varying scales from the two datasets presented by Li et al. (2023). Our analysis extends beyond previous studies by considering both pretrained and non-pretrained models (based on upstream language tasks), evaluating two-hop generation capabilities, and comparing models of varying sizes. To assess whether these models capture similar underlying game strategies, state representations, or other key aspects, despite differences in architecture and size, we employ representation alignment tools inspired by the literature on cross-lingual word embeddings (Søgaard et al., 2019). Finally, we visualize these results

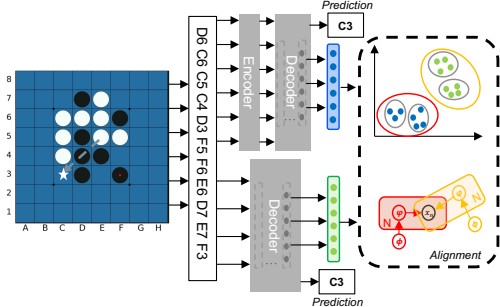

Figure 1: Experimental protocol. We re-train the Transformer-based models to predict the next move in Othello and see whether the board game layout is induced (up to isomorphism).

through latent move projections, enabling a deeper analysis of the internal mechanisms of models trained on the Othello game. Through these probing methods, we show that the language models – exhibit solid one-hop performance when trained on large amount of game sequence moves. We find that in some cases, all models can achieve up to 99% accuracy in unsupervised grounding, which means that absent any cross-modal supervision, a model *trained to play* Othello can identify the right positions on a board. More importantly, the alignment similarity score of the board features learned by these models is surprisingly high. Additionally, the latent move projection demonstrates that the models can learn the spatial structure of the chessboard. This provides a counter-example to previous claims that mono-modal models cannot solve visual question answering problems (Bender & Koller, 2020) – or, more generally, symbol grounding problems (Harnad, 1990). Beyond that, these results are significantly stronger than those in Li et al. (2023); Nanda et al. (2023b) and, in our view, provide more direct evidence of the Othello World Model Hypothesis[2].

## 2 RELATED WORK

**Past work on Othello** Most past works on Othello (Chang et al., 2018; van der Ree & Wiering, 2013) use reinforcement learning to search for moves. The first attempt to model Othello with deep neural networks dates back to 2018 (Liskowski et al., 2018), focusing on using CNNs to train a strong player. Based on it, Noever & Noever (2022) focus on designing an effective Othello player with LLMs. Motivated by Toshniwal et al. (2021), Li et al. (2023) shift the focus to treating the game as a diagnostic tool for inducing world models from text. Following this, Nanda et al. (2023b) provide evidence of a closely related linear representation of the board and propose a simple yet powerful way to interpret the model's internal state. Takizawa (2024) recently presents a provably optimal strategy for playing Othello, delving into the complexity of these strategies and paving the way for future research to explore whether LLMs adopt similar approaches. Hua et al. (2024) adopt the idea of Othello sequence generation and introduce a multilingual Othello task to aid in cross-lingual representation alignment.

**World models** The success of language models in NLP tasks, to many, seems to turn on their ability to simulate, predict, and reason about dynamic environments as portrayed in text (Hao et al., 2023; Huh et al., 2024; Patel & Pavlick, 2022; Xiang et al., 2023). The seminal work of Li et al. (2021) presents an example of fine-tuning LLMs on synthetic NLP tasks to find evidence that world states are weakly encoded in their activations. Wang et al. (2024) evaluate how well LLMs can serve as text-based world simulators with a benchmark. Inspired by Othello-GPT, research have explored more detailed probing (Yun et al., 2023; Hazineh et al., 2023) and more complex scenarios to assess

---

[2]Detailed definition see Appendix A.

| Method | Type | P | CHAMPIONSHIP | | | SYNTHETIC | | | | |
|---|---|---|---|---|---|---|---|---|---|---|
| | | | 2k | 20k | full | 2k | 20k | 200k | 2M | full |
| GPT-2 | D | ✗ | 49.8 | 17.7 | 5.6 | 49.2 | 26.8 | **13.6** | 10.4 | <0.1 |
| Bart | E-D | ✗ | 25.2 | 16.6 | 4.7 | 73.6 | 31.7 | 14.2 | 16.3 | <0.1 |
| T5 | E-D | ✗ | **20.9** | 15.2 | 4.3 | 65.8 | 28.7 | 15.7 | 10.1 | <0.1 |
| Flan-T5 | E-D | ✗ | 23.4 | **4.8** | **3.6** | **35.6** | **23.7** | 21.2 | **7.7** | <0.1 |
| LlaMa-2 | D | ✗ | 27.8 | 16.5 | 5.7 | 57.1 | 35.4 | 16.9 | 10.2 | <0.1 |
| Mistral | D | ✗ | 22.1 | 14.8 | 4.2 | 48.2 | 34.4 | 17.7 | 8.3 | <0.1 |
| Qwen2.5 | D | ✗ | 25.2 | 17.3 | 5.5 | 45.9 | 37.8 | 20.1 | 9.2 | <0.1 |
| GPT-2 | D | ✓ | 52.6 | 19.7 | 13.6 | 74.4 | 32.4 | 19.9 | 14.1 | <0.1 |
| Bart | E-D | ✓ | 54.0 | 14.6 | 13.7 | 77.2 | 35.8 | 24.4 | 16.6 | <0.1 |
| T5 | E-D | ✓ | 45.5 | 19.6 | 3.8 | 69.4 | 36.9 | 32.6 | 13.9 | <0.1 |
| Flan-T5 | E-D | ✓ | 31.7 | **4.8** | 3.7 | 70.3 | **25.4** | 45.0 | 8.7 | <0.1 |
| LlaMa-2 | D | ✓ | 43.1 | 14.7 | 7.0 | 74.6 | 41.5 | 33.4 | **7.6** | <0.1 |
| Mistral | D | ✓ | **16.8** | 15.0 | **3.3** | **33.8** | 30.6 | **18.2** | 7.7 | <0.1 |
| Qwen2.5 | D | ✓ | 20.9 | 18.2 | 6.0 | 46.5 | 39.3 | 23.4 | 10.8 | <0.1 |

Table 1: The error rate (%) of 1-hop game move generation in terms of different size of training data. 'Type' refers to the model type, 'P' denotes if the model is pretrained with upstream language modeling tasks or not. Numbers in bold represent best-performing models.

the ability of LLMs to understand board states, including for games like chess, checker and maze navigation (Karvonen, 2024; Joshi et al., 2024; Ivanitskiy et al., 2023). Our work aims to revisit the Othello World Hypothesis using novel probing methods across a number of different LLMs.

## 3 MODELING OTHELLO WITH LLMS

Following previous works (Liskowski et al., 2018; Li et al., 2023; Nanda et al., 2023b), we formulate the problem of playing the board game as a sequence generation problem. Specifically, we fine-tune generative pretrained models in an autoregressive manner to predict the next move given the current Othello board state. We then evaluate whether the predicted move is legal or not. Each game is a sequence, with each move represented as a token, and in each round, we predict the next move. Our vocabulary consists of 60 words, each representing one of the 60 playable tiles where players place discs, excluding the 4 center tiles, which are already occupied at the start of the game. See Figure 1 for an example move. Our modeling of Othello, in brief, can be represented as:

$$p_\theta(X_{i+k}|X_{<i}) = \prod_{m=0}^{k} p_\theta(X_{i+m}|X_{<i}) = \prod_{m=0}^{k} softmax(f_{i+m}(x_1, x_2, ..., x_{i+m-1})) \quad (1)$$

where $x_1, x_2, ..., x_{i-1}$ represent history moves, $X_{i+k}$ represents the sequence after $k$ generation steps. During inference, we input the previously generated game moves $X_{<i}$ at step $i$ into the model and prompt it to generate the next steps. Unlike previous works, we not only prompt the model to generate the next move ($k = 1$) but also introduce a new test where the model generates two consecutive moves ($k = 2$), for it prompts models to simulate high-level reasoning, revealing how well LLMs understand game strategy in a zero-shot manner.

### 3.1 EXPERIMENTAL SETUP

We use two datasets in our experiments, **CHAMPIONSHIP** and **SYNTHETIC**. Both of them were collected by Li et al. (2023). **CHAMPIONSHIP** comes from real online Othello gaming sources, whereas **SYNTHETIC** is artificially generated according to the rules of Othello game play. Detailed statistics see Appendix B. We use the last 20,000 games from each dataset for testing and validation (10,000 games each). Following Li et al. (2023), we report the top-1 error rate of the generated move. That if a generated move is not legal, we count it as an error. Specifically, we extend the 1-hop step generation setting in Li et al. (2023) and investigate 2-hop move generation for investigating the model's capability to anticipate more strategic, long-term planning in Othello. This involves

| Method | Type | P | CHAMPIONSHIP | | | SYNTHETIC | | | | |
|---|---|---|---|---|---|---|---|---|---|---|
| | | | 2k | 20k | full | 2k | 20k | 200k | 2M | full |
| GPT-2 | D | ✗ | 78.5 | 34.7 | 28.1 | 76.3 | 70.8 | **43.6** | 29.0 | 5.2 |
| Bart | E-D | ✗ | 54.2 | 31.1 | 23.4 | 86.5 | 67.2 | 44.8 | 35.7 | 4.2 |
| T5 | E-D | ✗ | **48.8** | 28.7 | 24.4 | 88.2 | 67.7 | 46.9 | 35.9 | 3.4 |
| Flan-T5 | E-D | ✗ | 51.8 | **20.8** | **21.9** | 79.6 | **63.1** | 48.6 | 26.7 | **2.8** |
| LlaMa-2 | D | ✗ | 60.9 | 36.3 | 26.4 | 87.3 | 67.8 | 45.2 | 36.3 | 5.5 |
| Mistral | D | ✗ | 51.4 | 31.7 | 22.3 | **71.2** | 77.1 | 47.9 | **26.4** | 3.0 |
| Qwen2.5 | D | ✗ | 55.9 | 25.4 | 22.8 | 77.6 | 65.3 | 44.2 | 28.7 | 3.3 |
| GPT-2 | D | ✓ | 92.2 | 43.4 | 37.2 | 99.6 | **72.6** | 45.5 | 34.4 | 6.2 |
| Bart | E-D | ✓ | 87.0 | 34.5 | 27.1 | 97.8 | 76.9 | 64.0 | 44.5 | 5.1 |
| T5 | E-D | ✓ | 86.5 | 36.4 | 27.0 | 99.6 | 78.8 | 59.9 | 46.9 | 4.6 |
| Flan-T5 | E-D | ✓ | 67.9 | **31.8** | 26.5 | 98.6 | 80.8 | 79.7 | 35.3 | 3.9 |
| LlaMa-2 | D | ✓ | 66.9 | 33.4 | 33.0 | 94.2 | 77.6 | 62.1 | **33.2** | 5.2 |
| Mistral | D | ✓ | **52.0** | 40.8 | **25.4** | **80.3** | 76.0 | **42.3** | 35.0 | **3.8** |
| Qwen2.5 | D | ✗ | 63.1 | 38.4 | 25.8 | 85.0 | 79.3 | 45.1 | 36.0 | 3.9 |

Table 2: The error rate (%) of 2-hop game move generation.

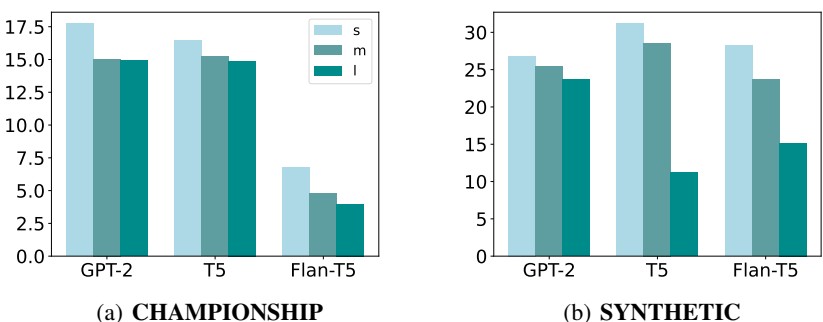

(a) **CHAMPIONSHIP**  (b) **SYNTHETIC**

Figure 2: Othello 1-hop generation error rate under different model sizes. All models are non-pretrained versions fine-tuned with 20k game sequences.

verifying whether the top-1 prediction is legal when the model is prompted to generate one and two moves at a time. We present the average error rate across all game sequences. We implement all of the baselines under the Pytorch framework and the HuggingFace model repository. We conduct all of our experiments using 8 A100 GPUs. We use all the default parameters in the repository when fine-tuning.

We perform our experiments using several existing baselines, with both Encoder-Decoder or Decoder-only structures. We first adopt some popular PLMs such as GPT-2 (Radford et al., 2019), T5 (Raffel et al., 2019), and Bart (Lewis et al., 2019). We then adopt several LLMs to see the their performance on this task, including Flan-T5 (Chung et al., 2022), LlaMa-2 (Touvron et al., 2023), Mistral (Jiang et al., 2023), and Qwen2.5 (Hui et al., 2024). Details see Appendix C.

## 3.2 EVALUATION OF LLM PERFORMANCE IN OTHELLO MOVE GENERATION

We perform experiments using various methods and present the results in Tables 1 and 2. From our observations, several key findings emerge. Firstly, there is **no clear winner** between models with an Encoder-Decoder architecture, such as T5 or Flan-T5, and those with a Decoder-only architecture, such as GPT-2, LLaMA-2, or Qwen2.5 in terms of performance on this task. This indicates that the architectural differences between these models do not significantly impact their ability to generate Othello game steps. However, one consistent trend is the positive correlation between the amount of training data and overall model performance. As we increase the scale of the training data, all mod-

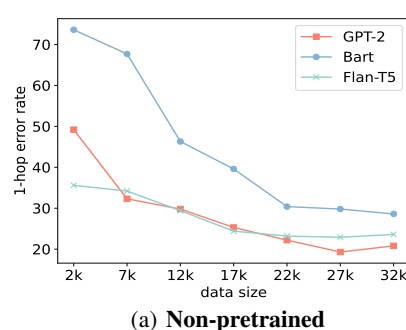 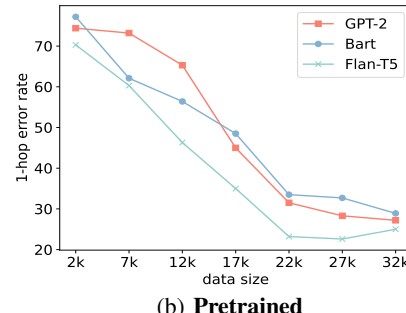

(a) **Non-pretrained**        (b) **Pretrained**

Figure 3: Analysis of 1-hop error rates on the SYNTHETIC dataset with varying data scales.

els tend to improve, underscoring the importance of data availability in mastering complex tasks like Othello move generation. In comparison to smaller language models, LLMs such as Mistral and Flan-T5 demonstrate clear superiority in this task, suggesting that **model size and capacity** are critical factors in understanding Othello game step generation. Larger models are better equipped to capture the intricate patterns and strategies within the game, likely due to their increased representational capacity. Interestingly, we also find that pretrained language knowledge, while generally beneficial for a wide range of natural language tasks, sometimes **negatively impacts** a model's ability to understand and generate game steps. Specifically, the pretrained versions of many models perform worse than their non-pretrained counterparts in this task, which could indicate that knowledge learned from upstream language tasks introduces biases or distracts from learning the specific structure and rules of Othello. Furthermore, while fine-tuning models on a large amount of data leads to reasonable performance in generating a single step (1-hop), generating more than one step consecutively remains a significant challenge. Even with large-scale data, models struggle to accurately predict two or more consecutive moves. This shows the potential limitation of the 1-hop evaluation since while it mostly focuses on the immediate next move based on the current board state, it inherently overlooks the deeper decision-making process required for gameplay strategies.

### 3.3 IMPACT OF MODEL SIZE ON OTHELLO MOVE GENERATION

To further explore the impact of model size on the ability to model Othello moves, we analyze the performance of various models across different size configurations, as depicted in Figure 2. For each model, we evaluate performance in small, medium, and large size versions, allowing us to compare how scaling up model capacity affects accuracy in generating game moves. The results show a clear trend: **as model size increases, the error rate consistently decreases across both datasets**. This trend is particularly pronounced in the SYNTHETIC dataset, where larger models achieve significantly lower error rates compared to their smaller counterparts. The stronger improvement in the SYNTHETIC dataset suggests that larger models are better at capturing the structured patterns present in the synthetic data, likely due to their enhanced capacity for learning complex representations and generalizing across more varied scenarios. These findings highlight the importance of model scaling, showing that increasing the model size can lead to substantial performance gains in Othello move generation, especially in environments where the data is highly structured or synthetic in nature. Furthermore, the results emphasize that larger models are not just marginally better, but often significantly outperform smaller models, reinforcing the need to consider model capacity as a critical factor when tackling tasks that require a deep understanding of game strategies and sequential decision-making processes.

### 3.4 IMPACT OF DATA SIZE ON OTHELLO MOVE GENERATION

In Table 2, we observe a sharp decrease in model error rates as the dataset size increases from 2k to 20k. To investigate this further, we conduct an analysis by gradually enlarging the SYNTHETIC dataset from 2k to 32k. According to Figure 3, the performance of all models improves gradually as the dataset size increases. Interestingly, non-pretrained models exhibit a faster reduction in er-

ror rates within the 2k to 12k data size range, with diminishing improvements beyond that point compared to pretrained models. In contrast, pretrained models take longer to achieve comparable performance, highlighting their slower adaptation to the task. This suggests that non-pretrained models are better suited for quickly learning game rules and adapting to fundamental patterns in the data. Furthermore, it indicates that the prior natural language knowledge embedded in pretrained models does not significantly enhance their understanding of the game. This observation aligns with our findings discussed in Section 3.2, where we also observed the limited impact of pretrained knowledge in tasks requiring specialized domain adaptation.

# 4 OTHELLO REPRESENTATION ALIGNMENT ACROSS LANGUAGE MODELS

Drawing inspiration from the literature on cross-lingual word embeddings, we perform Othello representation alignment across different models to compare how each model, despite differences in architecture and size, internalizes and represents game strategies and states. This helps us assess whether the learned representations in Section 3 are consistent across models and whether they capture similar underlying patterns essential for accurate Othello move generation.

## 4.1 ALIGNMENT METHOD

To validate the Othello World Model Hypothesis, we directly evaluate the internal representation of the Othello board in language models. Using the representations from different models, denoted as $F_1$, $F_2$ from the same input sequence $X_{<i}$, we perform mapping training under both *supervised* and *unsupervised* scenarios[3]. A linear mapping $W$ is learned to map $F_1$ and $F_2$ into the same space:

$$W^* = \underset{W \in M_i(\mathbb{R})}{\arg\min} ||WF_1 - F_2|| \tag{2}$$

where $F_1, F_2 \in \mathbb{R}^{i \times h}$ are representations from the final hidden Decoder layer in different language models trained for Othello generation. $M_i(\mathbb{R})$ is the space of $i \times i$ matrices of real numbers.

**Supervised training.** We consider the internal representations of different models within different source and target spaces. For supervised training (see Algorithm 1)[4], we use the pairwise data to learn a mapping from the source to the target space using iterative Procrustes alignment (Gower & Dijksterhuis, 2004). We use representations from two models as training pairs. Specifically, the representations of the $i$th step within the same game from the two models are considered a pair, denoted as $h_{\theta 1}(X_{<i})$ and $h_{\theta 2}(X_{<i})$, respectively. In our experiment, we randomly select 1,000 game sequences from the validation set as training pairs.

---

**Algorithm 1:** Supervised Training for Othello Representation Alignment

**Inputs :**
$h_{\theta 1}(\cdot), h_{\theta 2}(\cdot)$          representations from the final hidden layer of Decoder in two models: $\Theta_1, \Theta_2$
$X_{<i} = \{x_1, ..., x_{i-1}\}$   input game sequence at time step $i$
$r$                         number of refinement iterations

**Output:**
$s$     Similarity score of the aligned feature learned from the two models
$F_1 \leftarrow h_{\theta 1}(X_{<i}), F_2 \leftarrow h_{\theta 2}(X_{<i})$
**for** $i = 1$ *to* $r$ **do**
  **if** $i != 1$ **then**
    $F_1 \leftarrow \text{BuildDic}(F_1), F_2 \leftarrow \text{BuildDic}(F_2)$     `// build a dictionary from aligned`
    `embeddings containing best aligned pairs`
  $W \leftarrow \text{Procrustes}(F_1, F_2)$
  $F_1 \leftarrow WF_1$
$s \leftarrow CosSim(F_1, F_2)$

---

**Unsupervised training.** For unsupervised training, without any parallel data or anchor points, following Conneau et al. (2018), we learn the mapping through a combination of adversarial training

---

[3]Both of the algorithms are implemented using MUSE, a library designed for multilingual embedding alignment (https://github.com/facebookresearch/MUSE).

[4]More details (e.g. BuildDic() of Algorithms 1, 2) see Conneau et al. (2018).

and iterative Procrustes refinement (Lample et al., 2018) (see Algorithm 2). The process involves first learning an initial proxy of the mapping $W$ using an adversarial criterion. Where an additional Discriminator model is trained to identify the origin of an embedding, yet the target mapping $W$ aims at preventing the discriminator from doing so. Then, the mapping $W$ is further refined via Procrustes using the same strategy in supervised training. We then report the average cosine similarity of the aligned features on the test set.

---

**Algorithm 2:** Unsupervised Training for Othello Representation Alignment

**Inputs :**
| | |
|---|---|
| $h_{\theta 1}(\cdot), h_{\theta 2}(\cdot)$ | representations from the final hidden layer of Decoder in two models: $\Theta_1, \Theta_2$ |
| $X_{<i} = \{x_1, ..., x_{i-1}\}$ | input game sequence at time step $i$ |
| $k, r$ | number of adversarial training iterations, number of refinement iterations |

**Output:**
$\quad s \quad$ Similarity score of the aligned feature learned from the two models
$F_1 \leftarrow h_{\theta 1}(X_{<i}), F_2 \leftarrow h_{\theta 2}(X_{<i})$
$\text{RandomInitialize}(W)$
**for** $i = 1$ *to* $k$ **do**
$\quad \mathcal{D} \leftarrow \text{TrainDiscriminator}(\mathcal{W}, \mathcal{D}, F_1, F_2)$        `// train the discriminator` $\mathcal{D}$
$\quad W \leftarrow \text{FoolDiscriminator}(\mathcal{W}, \mathcal{D}, F_1, F_2)$   `// train` $W$ `to fool the discriminator`
$F_1 \leftarrow W F_1$
**for** $i = 1$ *to* $r$                                               `// refine` $W$
**do**
$\quad F_1 \leftarrow \text{BuildDic}(F_1), F_2 \leftarrow \text{BuildDic}(F_2)$
$\quad W \leftarrow \text{Procrustes}(F_1, F_2)$
$\quad F_1 \leftarrow W F_1$
$s \leftarrow CosSim(F_1, F_2)$

---

## 4.2 MAPPING RESULT

We probe different models by aligning their representations into one joint vector space. We report the cosine similarity of the aligned features score under both supervised (Conneau et al., 2018) and unsupervised (Lample et al., 2018) settings in Table 3[5].

| Src. | Trg. | Supervised | | Unsupervised | |
|---|---|---|---|---|---|
| | | CHAM. | SYN. | CHAM. | SYN. |
| GPT-2 | Bart | 81.4 | **93.1** | 80.3 | 91.3 |
| GPT-2 | T5 | 83.0 | 85.0 | 76.4 | 80.1 |
| Bart | T5 | 69.2 | 84.5 | 85.2 | 81.1 |
| GPT-2 | Mistral | **90.3** | 77.2 | 80.3 | 82.6 |
| Bart | Mistral | 88.0 | 79.1 | **96.1** | **97.2** |
| LlaMa-2 | Mistral | 80.1 | 74.2 | 76.2 | 72.6 |
| Qwen2.5 | LlaMa-2 | 84.2 | 80.1 | 81.3 | 84.9 |

Table 3: Representation alignment cosine similarity (%) results. Src. and Trg. represent source and target space. CHAM., SYN represent CHAMPIONSHIP and SYNTHETIC dataset.

From the results, we observe consistently high similarity scores across different language models, indicating that despite architectural differences, these models capture similar underlying representations when tasked with the Othello game. For instance, the SYNTHETIC supervised similarity score between GPT-2 (a Decoder-only model) and Bart (an Encoder-Decoder model) reaches an impressive 93.1%. This suggests that, although these models process information differently due to their structural variances, they still converge on shared knowledge and representations when learning to model the Othello task. Such a high similarity score points to the possibility that both model types learn similar strategic patterns and rules intrinsic to the game, reinforcing the idea that fundamental aspects of the Othello task are captured across architectures.

---

[5]We use the non-pretrained version based on 20k training data for all models.

## 4.3 PCA Visualization

In order to vividly show such alignment, we also demonstrate the dimension-reduced PCA coordinate of 60 step features $h_\theta(X)$ within one entire random game in Figure 4. We also observe highly similar step representations across different models. This suggests that these models are learning comparable internal representations of the game states, indicating that the models are aligned in how they interpret the sequential nature of Othello. Even though they may be built differently (e.g., Decoder-only versus Encoder-Decoder), the core representations they learn about the game states converge to a similar space.

This result highlights a level of consistency and robustness in the way generative models process game-related information. Despite differences in architecture or training objectives, the models seem to internalize and represent Othello game states in a similar manner. This convergence suggests that these models, when trained on the Othello task, are not only learning task-specific patterns but are also aligning on a shared understanding of the underlying problem space. To sum up, such alignment enhances the interpretability of these models, as their internal representations become more comparable.

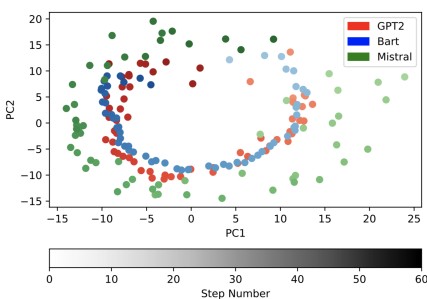

Figure 4: PCA visualization of the 60 steps from various models within one game.

## 4.4 Mapping Across Different Layers

We compare the mapping similarity across different Decoder hidden layers in GPT-2 and Flan-T5[6] to understand how each model progressively learns to represent the Othello game, evolving from simple board states to more complex strategies. As shown in Figure 5, despite their structural differences, GPT-2 and Flan-T5 exhibit similar learned representations at corresponding layers. Both models, when trained on Othello game sequences, seem to converge toward learning comparable internal representations, as highlighted by the heatmap. This conclusion is supported by the following observations:

(1) **High Similarity in the Upper Right Diagonal**. The heatmap reveals a prominent diagonal pattern where corresponding layers from GPT-2 and Flan-T5 show high similarity, especially in the upper half of the heatmap. This suggests that, despite their differing architectures (GPT-2 being autoregressive and Flan-T5 following an Encoder-Decoder structure), models eventually learn something in common (particularly at layer 11, where high similarity is observed) despite the difference from the beginning. This alignment indicates that their layerwise learning processes evolve in comparable ways as they both adapt to the Othello game environment. (2) **Layer-Specific Correspondences**. We notice that specific layers in GPT-2 show high similarity with certain layers in Flan-T5, even though they may not follow a strict diagonal pattern, this suggests that both models are learning certain shared features or patterns in game sequences at particular stages of their processing pipelines.

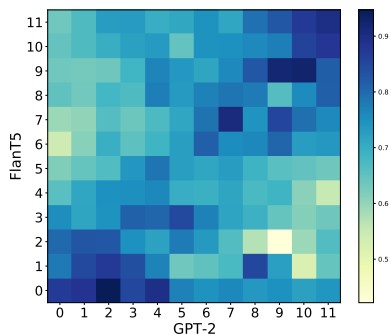

Figure 5: Decoder feature similarity heatmap across different layers.

---

[6]We use GPT-2-small and Flan-T5-Base trained on 20k SYNTHETIC dataset, as both have 12 decoder hidden layers.

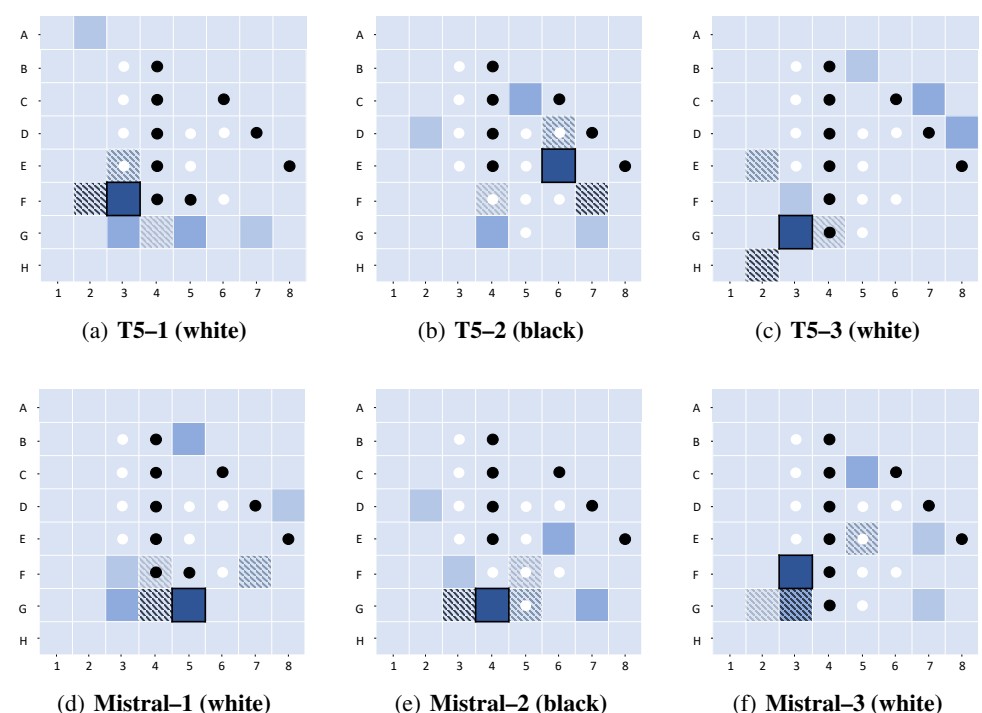

(a) **T5–1 (white)**  (b) **T5–2 (black)**  (c) **T5–3 (white)**

(d) **Mistral–1 (white)**  (e) **Mistral–2 (black)**  (f) **Mistral–3 (white)**

Figure 6: Othello latent move projection from two best performed models. Colors indicate the likelihood of the position of the next step. Shadows highlight the top three tiles with embeddings closest to the top candidate, with the darkest color in the black box.

## 5 LATENT MOVE PROJECTION: WHAT ELSE DOES LLMS LEARN?

To gain deeper insights into how models learn strategies and predict future moves, we project latent features onto a visual space. For a given game sequence $X_{<i}$, we highlight the top-5 candidate tile positions with the highest predicted probabilities for the next move. Additionally, we compare the embeddings of the top candidate tile with those of the other tiles. We mark the top three tiles whose embeddings are closest to the top candidate to examine their spatial relationships on the board.

We perform latent move projection on the Othello game steps of two models in Figure 6 (results for other models in Appendix G). It shows that both models successfully predict legal moves given a game sequence. Moreover, other legal moves are also assigned high prediction scores (tiles with lighter blue) by the models. This proves that with a large amount of game sequence data, the model learns the rule of the game. To further investigate whether the models can capture the physical position of each tile, we use shadow marks to highlight the tiles with the closest embedding distance to the tile in the black box. The intensity of the shadow reflects the degree of similarity. We observe that the top-1 tile with the highest similarity (F2 in Figure 10(a), G4 in Figure 10(g)) is the one adjacent to the black box tile in both models. This indicates that the models not only understand game mechanics but also capture the spatial relationships between tiles.

## 6 LIMITATIONS

Although this work demonstrates the ability of different language models to understand Othello game rules, several limitations persist that require further investigation:

**Challenges in Multi-step Move Generation.** While language models can predict the next move with reasonable accuracy, they struggle to predict entire game sequences. The key question is whether strong multi-step performance is a reasonable expectation. Othello is a dynamic game where optimal play often involves sacrificing short-term gains for long-term advantages. The com-

plexity arises from the interplay of distinct player strategies and the rotational invariance of the board, leading to many game states where the subjectively or objectively best move is inherently underdetermined. As a result, the ability to accurately predict entire sequences may remain elusive, given the complexity and variability of decision-making in the game.

**Limitations in Data Requirements.** Our experiments show that reducing the 1-hop error rate to less than 0.1% demands a large volume of training data. This reliance on vast datasets presents a scalability issue, as access to Othello game data is limited. Moreover, training on such large datasets is computationally expensive and time-consuming, which can be a prohibitive factor for many researchers or organizations without access to substantial computational resources.

## 7 POTENTIAL IMPACT

Training language models on Othello game sequences can imply that LLMs function as a world model because it showcases their ability to learn and internalize the structured dynamics and rules of a complex system, rather than merely memorizing patterns. Investigating the parallels between how language models learn structured representations and how humans internalize similar concepts can shed light on the cognitive processes underlying reasoning, strategy, and language. This could deepen our understanding of human cognition and inform theories of learning and representation. The observation that language models, regardless of architecture or scale, learn similar patterns from Othello game sequences suggests that these models converge on universal representations when trained on structured data. This implies that the underlying mechanisms of representation learning in LLMs are robust and consistent, highlighting their ability to capture the rules and dynamics of structured systems. The ability of language models to learn patterns from Othello sequences provides more hints on the idea that LLMs can act as world models, capable of internalizing rules, strategies, and dynamics. This has far-reaching implications for tasks requiring reasoning about complex environments, such as planning, simulation, and autonomous decision-making.

## 8 FUTURE DIRECTIONS

We list several possible future directions to study how our results could generalize to other broader scenarios.

**More Complicated Games.** Since this work is primarily limited to the Othello game, an intriguing question arises: could similar findings be observed in other games such as chess, checkers, or Go? These games, like Othello, involve strategic planning, dynamic state transitions, and trade-offs between short-term gains and long-term advantages. Exploring how large language models (LLMs) learn and represent strategies in these contexts could be highly valuable.

**Multimodal Support.** Leveraging Multimodal LLMs (MLLMs) to train models and investigate feature alignment across different modalities is also a highly relevant and promising research direction. In the context of Othello, this approach could involve aligning visual representations of the game board with text-based sequencial moves. Such alignment can help bridge the gap between symbolic reasoning and natural language understanding, enabling models to not only predict optimal moves but also provide hints if the world model theory could also be applied in other modalities.

## 9 CONCLUSION

We conduct a detailed probing of language models' ability to predict legal moves in the Othello board game, based on the settings in Li et al. (2023). We evaluate seven language models, training them to predict the next move based on previous moves. All seven models achieve almost 'perfect' one-hop move prediction performance when trained with large amount of data. We then adopt representation alignment tools to align the learned game state features from different models into one joint space. We observe high similarity in the board features they learned. In addition, latent move projection is performed to show the models not only understand the game mechanics but also capture the spatial relationships between tiles. These results, in our view, provide more solid evidence to date of the Othello World Model Hypothesis presented in previous works.

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

## A  OTHELLO WORLD MODEL HYPOTHESIS

According to previous works Li et al. (2023); Nanda et al. (2023a), a world model refers to a representation or a mapping of a world, ideally a homomorphism. Language models have been shown

| | CHAMPIONSHIP | SYNTHETIC |
|---|---|---|
| Num. of Games | 132,588 | 23,796,010 |
| Avg. length | $59.8 \pm 1.5$ | $60.0 \pm 0.8$ |
| Min. length | 4 | 9 |
| Full length portion(%) | 95.0 | 99.1 |

Table 4: Dataset statistics of the two Othello datasets.

to develop internal representations for simple concepts, such as color and direction Patel & Pavlick (2022); Abdou et al. (2021). Training language models on Othello game sequences further supports the idea that LLMs can function as a world model. This is demonstrated by their ability to learn and internalize the structured dynamics and rules of a complex system, rather than merely memorizing patterns. This capacity highlights their potential for understanding and representing intricate environments through abstract, systematic reasoning.

## B    DATASET STATISTICS

The details of the two datasets are listed in Table 4.

## C    COMPARED METHODS

We perform our experiments using several existing baselines, with both Encoder-Decoder or Decoder-only structures. We first adopt some popular language models such as

**GPT-2.** We fine-tune GPT-2 to generate the whole game sequence step by step. Specifically, we use the smallest version of GPT-2.

**Bart.** We use Bart-base to generate the sequence by feeding the first token into the Encoder and fine-tuning the model to generate the remaining tokens.

**T5.** Similar as Bart, we adopt `T5-base` in our experiment.

We then adopt several LLMs for the task:

**Flan-T5.** We adopt `Flan-T5-XL`, which contains 3B parameters in our experiment.

**LLaMA-2.** We use `LlaMa2-7B` and only fine-tune the LoRA adapter in our experiment.

**Mistral.** We use `Mistral-7B` in our experiments. Similar to LLaMA-2, we also only fine-tune the LoRA adapter but keep the rest of parameters fixed.

**Qwen2.5.** We use `Qwen2.5-7B` in our experiments, one of the most state-of-the-art LLMs for sequence generation.

## D    MODEL SIZE ANALYSIS ON TWO-HOP GENERATION

We present the 2-hop performance across various model sizes in Figure 7. As we scale up the model, the error rate decreases, suggesting that a larger model size positively affects game understanding. However, the impact of model size diminishes when compared to the 1-hop performance, indicating a diminishing return on performance gains with increased model size.

## E    DATA SIZE ANALYSIS ON CHAMPIONSHIP DATASET

We also present the data size analysis on the CHAMPIONSHIP dataset (see Figure 8). We see similar conclusions as in Figure 3. The prediction accuracy gets better when we increase the data size. Also, the error rate demonstrates a more steady drop in models pretrained with upstream language modeling tasks.

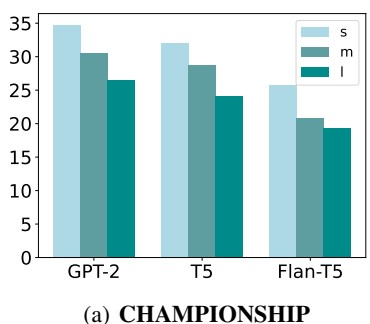 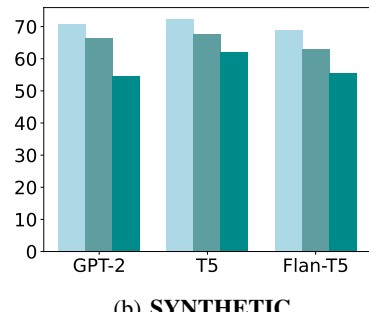

(a) **CHAMPIONSHIP**    (b) **SYNTHETIC**

Figure 7: Othello 2-hop generation performance under different model sizes. All models are non-pretrained versions fine-tuned with 20k game sequences.

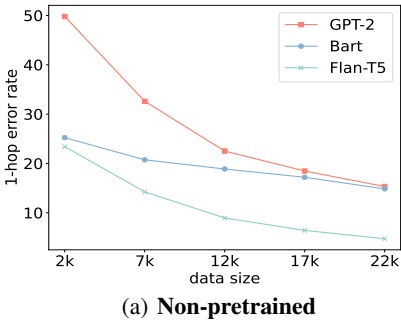 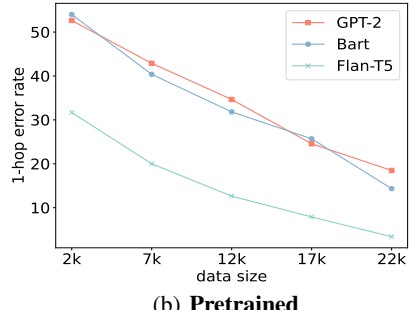

(a) **Non-pretrained**    (b) **Pretrained**

Figure 8: Analysis of 1-hop error rates on the CHAMPIONSHIP dataset with varying data scales.

## F  SUPERVISED MAPPING HEATMAP

We also present the supervised mapping results for the same setting in Section 4.4. The mapping in Figure 9 reveals a more pronounced diagonal pattern of similarity, with particularly high similarity observed in the upper-right corner. This provides further evidence that the models converge and acquire shared knowledge when trained on Othello data, indicating a strong alignment in their learned representations.

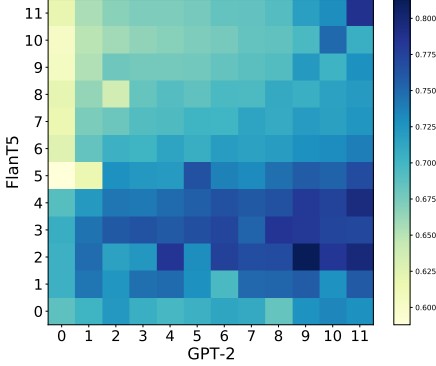

Figure 9: Decoder feature similarity (supervised) heatmap across different layers.

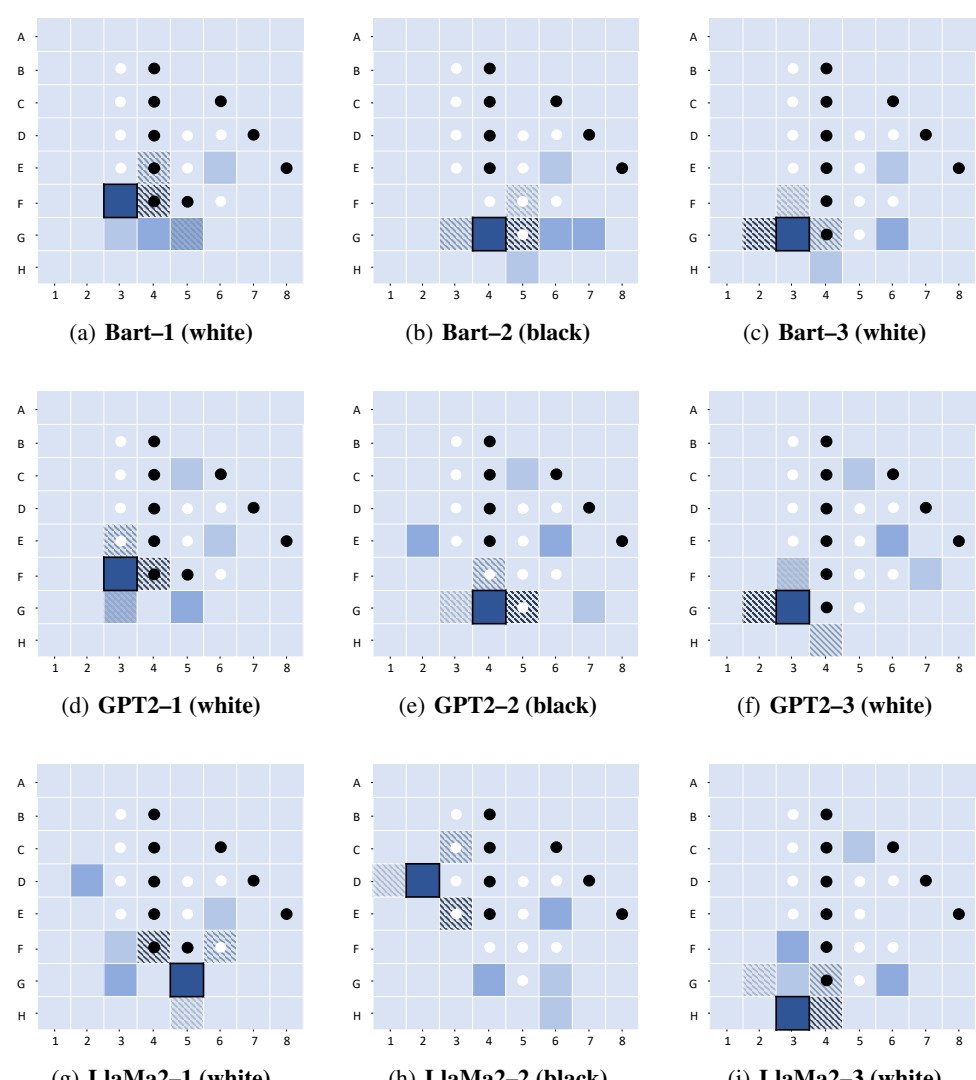

Figure 10: Othello latent move projection from two best performed models.

# G LATENT MOVE PROJECTION (FULL VERSION)

We attach the prediction from different models of the same game state in Figure 10. By comparing the performance of different models on the task, we find that overall, Mistral shows the best performance. It consistently demonstrates the best performance across different scenarios, effectively generating legal moves and showing a nuanced understanding of game rules. The Bart model frequently predicts adjacent tiles, leading to numerous failure cases, particularly when trained with smaller datasets. Llama-2 exhibits inconsistent performance, with a tendency to favor certain tile positions or exhibit a bias in move selection. While its predictions are often reasonable, the model appears to lack the robust policy understanding seen in Mistral, especially under constrained training conditions.

