, pretrained models demonstrate a more consistent and steady decrease in error rate compared to non-pretrained models. This suggests that pretrained models are able to effectively utilize the additional data to refine their representations and decision-making processes in a more stable manner, benefiting incrementally from larger datasets. In contrast, non-pretrained models show a more pronounced reduction in error rates within the 2k to 12k data size range, with diminishing improvements beyond that point. This indicates that while non-pretrained models experience substantial early gains from additional training data, their performance plateaus at larger dataset sizes, likely because they have already captured the most critical game strategies from the smaller datasets. The divergent behavior between pretrained and non-pretrained models suggests that pretraining on upstream natural language tasks enables models to leverage larger datasets for gradual improvements, whereas non-pretrained models rely more heavily on immediate learning from the provided game-specific data. The initial sharp gains for non-pretrained models also imply that these models are more sensitive to smaller datasets, rapidly improving as they acquire game-specific knowledge but requiring increasingly larger datasets for further marginal gains.

# 4 OTHELLO REPRESENTATION ALIGNMENT ACROSS LANGUAGE MODELS