# OpenReview forum: "Revisiting the Othello World Model Hypothesis"
_ICLR.cc/2025/Conference — Submitted to ICLR 2025_

### Official Review · Reviewer_duNp · 2024-10-30

**Soundness:** 3
**Presentation:** 3
**Contribution:** 3
**Rating:** 5
**Confidence:** 4

**Summary:**

In 2023 Li et al. (and subsequently Nanda et al. (2023)) formulated the Othello World Model Hypothesis (OWRH), claiming that GPT-2, based purely on Othello move sequence analysis, was able to infer the principles of the game, including its 64-square board representation. This paper revisits OWRH with 6 Large Language Models (LLMs) and enhanced research protocol, providing stronger evidence supporting the hypothesis than the two above-cited articles.

**Strengths:**

1) The paper is clearly written and easy to follow.
2) The experiments are well-thought and lead to several new insights.
3) The topic should be of interest to some of the ICLR community.

**Weaknesses:**

1) The novelty of the paper is limited. The underlying research concept of verifying the OWMH is not new and even though the paper leads to certain new observations, they are not surprising and do not significantly expand the existing knowledge.
2) The selection of LLMs is somewhat outdated, since there are quite a few stronger LLMs available these days.
3) In the era of MLLMs (Multimodal LLMs) the rationale behind the proposed research is disputable.

**Questions:**

1) What qualitatively new observations related to the internal representation of Othello games in LLMs result from the presented study? What are the high-level novel implications of the presented experiments and conclusions?
2) Why this particular set of models has been selected? There are quite a few newer models available at the moment, both proprietary and open access.
3) How the presented study relates to the representation abilities of MLLMs?

---

> ### Author Response · Authors · 2024-11-22
> **Official Comment by Authors**
>
> **Response to weakness**
> > 1. The novelty of the paper is limited. The underlying research concept of verifying the OWMH is not new and even though the paper leads to certain new observations, they are not surprising and do not significantly expand the existing knowledge.
>
> We respectfully disagree. We show very clearly what the limitations of previous work were, and are first to show that a world model of Othello emerges from playing Othello. Our methodology (and, as a result, evidence) is much stronger than previous works.
>
> > 2. The selection of LLMs is somewhat outdated, since there are quite a few stronger LLMs available these days.
>
> We appreciate the reviewer’s feedback regarding the selection of LLMs. While we recognize that newer and potentially stronger LLMs have become available, the selected models, such as GPT-2, T5, and Bart, LlaMa2, are well-documented, widely used, and readily available to the research community. By using these models, we aim to ensure our findings are easily reproducible. **We argue that the main focus of this paper should not be adopting state-of-art methods for obtaining SOTA performance, but more on investigating whether increasing model capacity could truly help understand the Othello game.** However, we’ve added some more recent LLMs (e.g. Qwen 2.5) in Table 1, 2  in our newest version and also report the performance of LLaMA 3.1 below to address the reviewer’s concern.
>
>
> > 3. In the era of MLLMs (Multimodal LLMs) the rationale behind the proposed research is disputable.
>
> We thank the reviewer for highlighting this important point. We acknowledge that leveraging Multimodal LLMs (MLLMs) to train models and investigate feature alignment across different modalities is a highly relevant and promising research direction. In fact, we have identified this as an essential avenue for future exploration and **have already initiated work in this area**. However, as this direction extends beyond the scope of the current paper, we have chosen to pursue it as a separate line of work. *To clarify this, we explicitly discuss it in the Future Work section of the newest version of our paper.*
>
>
>
> *Response to Questions**
> > 1. What qualitatively new observations related to the internal representation of Othello games in LLMs result from the presented study? What are the high-level novel implications of the presented experiments and conclusions?
>
> Kenneth Li and colleagues initially sought to demonstrate that LLMs acquire semantics rather than being "syntax all the way down." However, their study is limited to small-scale language models, specifically GPT-2, leaving open several important questions. For instance, it remains unclear whether their findings generalize to larger-scale language models or how much training data is required to achieve "perfect" performance. Additionally, their study does not explore whether differences in model architecture could yield similar levels of game understanding. More broadly, we extend this line of inquiry by probing whether language models understand the game's strategy or merely its rules. To address this, we train models to generate sequences comprising multiple moves at a time, pushing beyond simple rule-based learning. Our experiments reveal that different language models, regardless of their architecture, exhibit high similarity in the learned features. This finding provides additional support for the Othello world model theory, suggesting that language models can internalize representations of game rules and strategies through exposure to simple game sequences.
>
>
>
> > 2.Why this particular set of models has been selected? There are quite a few newer models available at the moment, both proprietary and open access.
>
> Please see our response for weakness 2.
>
> > 3.How the presented study relates to the representation abilities of MLLMs?
>
> Please see our response for weakness 3.

---

> > ### Author Response · Authors · 2024-11-22
> > **Official Comment by Authors (1)**
> >
> > *Table 1: 1-hop game move generation error rate of different models. (c) denotes the CHAMPIONSHIP data and (s) denotes the SYNTHETIC data.*
> > | Method      | 2k (c)    | 20k (c)    | 200k (c)    |  |  2k (s)    | 20k (s)    | 200k (s)    | 2M (s)    | full (s)    |
> > |-------------|-------|-------|-------|------|-------|-------|------|-------|-------|
> > |Qwen 2.5 (non-pretrained) | 25.2|17.3|5.5||45.9|37.8|20.1|9.2|<0.1|
> > |LlaMa 3.1 (non-pretrained) |21.7|9.3|3.8| |37.1|25.5|13.9|8.2|<0.1|
> > |-------------|-------|-------|-------|------|-------|-------|------|-------|-------|
> > |Qwen 2.5 (pretrained) | 20.9|18.2|6.0||46.5 |39.3|23.4|10.8|<0.1|
> > |LlaMa 3.1 (pretrained) |23.8|11.2|4.1||39.3|26.6|21.5|8.9|<0.1|
> >
> > *Table 2: 2-hop game move generation error rate of different models.*
> > | Method      | 2k (c)    | 20k (c)    | 200k (c)    |  |  2k (s)    | 20k (s)    | 200k (s)    | 2M (s)    | full (s)    |
> > |-------------|-------|-------|-------|------|-------|-------|------|-------|-------|
> > |Qwen 2.5 (non-pretrained) | 55.9|25.4|22.8||77.6|65.3|44.2|28.7|3.3|
> > |LlaMa 3.1 (nonpretrained) |50.6|22.7|21.2||73.4|64.8|44.0|26.9|3.0|
> > |-------------|-------|-------|-------|------|-------|-------|------|-------|-------|
> > |Qwen 2.5 (pretrained) | 63.1|38.4|25.8||79.3 |65.3|45.1|36.0|3.9|
> > |LlaMa 3.1 (pretrained) | 58.2|34.1|25.5||82.6|75.4|45.8|34.9|4.0|
> >
> > We thank the reviewer again for the suggestions. We've added more elaborations and experimental results concerning the problems discussed in our newest version. We sincerely hope the reviewer can consider these revisions during the rebuttal phase and kindly reassess the overall score.

---

> > > ### Comment · Reviewer_duNp · 2024-11-27
> > >
> > > I'd like to thank the authors for the rebuttal, which cleared up some of my doubts, though I still believe the contribution is incremental.
> > >
> > > I'll keep my original rating.

---

### Official Review · Reviewer_ezgd · 2024-11-02

**Soundness:** 2
**Presentation:** 1
**Contribution:** 2
**Rating:** 3
**Confidence:** 4

**Summary:**

This paper aims to add additional evidence to the Othello World Model Hypothesis by training a variety of different LLMs for predictive tasks in the game of Othello.

**Strengths:**

Highly topical area of research, and I think the experiments are carried out well.

**Weaknesses:**

Under strengths above, I wrote that I **think** experiments are carried out well. Here is my main issue with the paper: many crucial details are not described in sufficient detail, and/or are too vague. I can make reasonable guesses as to the exact work that was done, and based on this, I do genuinely believe there is good work in here. But I shouldn't have to guess, and this state is not acceptable for a research paper.

I will elaborate on some specific points:
1. The entire paper revolves around the hypothesis that LLMs trained on Othello move sequences can induce a "relevant world model". But... I'm missing a definition of world model. I cannot judge with full certainty whether the experiments adequately support the claims, when I don't even have a crisp, clear, unambiguous definition of the hypothesis that the entire paper revolves around. I understand that "world model" is a relatively common phrase, but it is still crucial to define it clearly and unabmiguously.
2. The paper does not make it clear exactly what the models are trained to do. Combined with the lacking definition of world model above, this makes things very problematic. It is not clear to me whether the models are trained to:

    - Given current state (implied by sequence of previous moves), predict what the next played move is / should be.
    - Given current state (implied by sequence of previous moves) and a next move, predict what the next state will be.
    - A combination of the above, or anything else.

The caption of Figure 1 talks about "predict the next move". The caption of Table 1 is talking about "game state generation". These two are two very different things. Much of the rest of the paper talks about "move generation", which could be predicting next move again, but could also be about predicting which moves are legal, for instance.

3. There are no details whatsoever on how the SYNTHETIC dataset was generated. Which agents were used to play these games? This requires complete details on these agents (what algorithms, how much search time if they used search, on what hardware, any kind of randomisation used to ensure variety in the data, ... we need to know everything, but now we know nothing at all).

Other comments:
- Section 2 says that the work of Takizawa (2024) also looked at "whether LLMs adopt similar ones [strategies]", but as far as I can see, they did not do anything even remotely like that at all.
- line 159 PLMs should be LLMs?
- Section 3.2 refers to Tables 2 and 3, but this should be 1 and 2?
- Caption of Figure 2 vaguely mentions "performance". This is not precise enough (could be accuracy, could be error rate, would lead to very different interpretations). There's also no label on the y-axis, which also does not help in this regard.
- Line 263/264 talks about performance plateauing, but I don't see it as plateauing at all. Therefore, I also disagree with much of the analysis in the rest of the bottom of page 5. Sure, the decline in error rate becomes less steep at the end for the non-pretrained models. But they didn't fully plateau yet, and are **still** outperforming the Pretrained models **also at the very end of your x-axis**. These observations disagree with much of your conclusions here.
- Line 462/463 mentions "the policy of the game". There is no such thing as a "the policy" of any game. We can play according to many different policies.

**Questions:**

1. Please define world model.
2. Please describe very precisely what the models are actually trained to do.
3. Please provide details on how the SYNTHETIC dataset was generated exactly.

---

> ### Author Response · Authors · 2024-11-22
> **Official Comment by Authors**
>
> **Response to weakness**
> > 1. Section 2 says that the work of Takizawa (2024) also looked at "whether LLMs adopt similar ones [strategies]", but as far as I can see, they did not do anything even remotely like that at all.
>
> We apologize for the mistake. What we want to mean here is that ‘paving the way for future research to explore whether LLMs adopt similar approaches’. We’ve made the correction in our newest version.
>
> > 2.line 159 PLMs should be LLMs?
>
> We use the term ‘PLMs’ to refer to ‘pretrained language models,’ representing smaller-scale models such as GPT-2 and T5. This distinction is made to differentiate them from large language models (LLMs) like LLaMA and Flan-T5.
>
> > 3.Section 3.2 refers to Tables 2 and 3, but this should be 1 and 2?
>
> Thank you for pointing the typo out. We’ve corrected this typo in our newest version.
>
> > 4.Caption of Figure 2 vaguely mentions "performance". This is not precise enough (could be accuracy, could be error rate, would lead to very different interpretations). There's also no label on the y-axis, which also does not help in this regard.
>
> We apologize for not making this clear. Same as Table 2, the performance refers to error rate performance. We’ve changed the caption in our newest version.
>
> > 5.Line 263/264 talks about performance plateauing, but I don't see it as plateauing at all. Therefore, I also disagree with much of the analysis in the rest of the bottom of page 5. Sure, the decline in error rate becomes less steep at the end for the non-pretrained models. But they didn't fully plateau yet, and are still outperforming the Pretrained models also at the very end of your x-axis. These observations disagree with much of your conclusions here.
>
> We clarify that non-pretrained models plateau in performance when the data size increases from 22k to 27k and 32k, which is not shown in the figure. In the figure, it is evident that the performance of non-pretrained models, such as GPT-2 and Flan-T5, remains less changed when increasing the data size from 12k to 22k. We have updated the corresponding analysis text to make it clearer. While Figure 3 does not explicitly show plateauing for non-pretrained models—indicating instead a slow improvement near the end of the x-axis—we kindly argue that our claims about the comparative trends between pretrained and non-pretrained models remain valid. Specifically, non-pretrained models exhibit sharp, intermediate performance gains on smaller datasets, whereas pretrained models show a more gradual improvement as data size increases.
>
> > 6.Line 462/463 mentions "the policy of the game". There is no such thing as a "the policy" of any game. We can play according to many different policies.
>
> We apologize for our lack of precision. This should be the rules of the game - or similar.

---

> > ### Author Response · Authors · 2024-11-22
> > **Official Comment by Authors (1)**
> >
> > **Response to Questions**
> > > 1. Please define world model.
> >
> > A world model is a global theory of the world. A water with a hole in it can be a water clock, and while the bucket’s interior can be said to be in a modeling relationship with time, the bucket is not a world model. It is a model of something very local. Training language models on Othello game sequences can imply that LLMs function as a world model because it showcases their ability to learn and internalize the structured dynamics and rules of a complex system, rather than merely memorizing patterns.
> >
> > > 2. Please describe very precisely what the models are actually trained to do.
> >
> > We apologize for any confusion regarding the objective our models are trained to achieve. To clarify, the models are trained to predict the next move in a sequence, given the preceding moves. For evaluation, we measure the proportion of predicted moves that are legal within the context of the game. This approach follows the problem setting established in previous work [1]. For example, given the sequence of previous moves ‘D6C6C5,’ the model is expected to predict a move like ‘C4’ for evaluation. We have revised the caption for Table 1 and improved the description in Section 3 to ensure clarity and avoid further misunderstandings.
> >
> > > 3. Please provide details on how the SYNTHETIC dataset was generated exactly.
> >
> > We would like to clarify that we did not create the SYNTHETIC dataset on our own but instead utilized the existing SYNTHETIC dataset provided in [1]. As a result, we did not include extensive details about its construction in our paper, assuming that readers could refer to the original work [1] for more information. According to [1], the SYNTHETIC dataset was generated by uniformly sampling leaf nodes from the Othello game tree. This results in a data distribution that differs significantly from championship games, as it does not reflect any strategic considerations.
> >
> > *[1] Li et al. Emergent World Representations: Exploring a Sequence Model Trained on a Synthetic Task.*
> >
> > We thank the reviewer again for the suggestions. **We've created the corresponding parts in the paper and added more elaborations and experimental results concerning the problems discussed in our newest version.** We sincerely hope the reviewer can consider these revisions during the rebuttal phase and kindly reassess the overall score.

---

> > > ### Comment · Reviewer_ezgd · 2024-11-22
> > >
> > > Thank you for your responses.
> > >
> > > ---
> > >
> > > **On the discussions of Figure 3:**
> > >
> > > > We clarify that non-pretrained models plateau in performance when the data size increases from 22k to 27k and 32k, which is not shown in the figure.
> > >
> > > If it's not shown in the figure, I can't see it, and I also can't agree with any conclusions that are derived from it. If you do have data on this, please just include it in the figure.
> > >
> > > > In the figure, it is evident that the performance of non-pretrained models, such as GPT-2 and Flan-T5, remains less changed when increasing the data size from 12k to 22k.
> > >
> > > Yes, but rate of change in isolation is not important, not when the starting points are so different. The way I read the data in these figures is: the non-pretrained models already achieve good performance (possibly getting close to saturating?) much more quickly. They simply become slower later on, because they are already close to as good as they can be. The pretrained models only show higher rates of changes in the later changes, because they are simply slower throughout the whole trajectory, so they can look "faster" in the later stages because they still have to catch up and didn't saturate yet. You currently stop the $x$-axis right before the point that we actually need to see.
> > >
> > > > While Figure 3 does not explicitly show plateauing for non-pretrained models—indicating instead a slow improvement near the end of the x-axis—we kindly argue that our claims about the comparative trends between pretrained and non-pretrained models remain valid. Specifically, non-pretrained models exhibit sharp, intermediate performance gains on smaller datasets, whereas pretrained models show a more gradual improvement as data size increases.
> > >
> > > I still don't agree, or at least not with the tone of phrasing. I see not a single point on the $x$-axis in your figure where pretrained models are better than non-pretrained ones, yet much of the text in the paper makes the story sound positive for the pretrained models.
> > >
> > > ---
> > >
> > > > We apologize for our lack of precision. This should be the rules of the game - or similar.
> > >
> > > This was not yet updated in the PDF.
> > >
> > > ---
> > >
> > > > A world model is a global theory of the world. A water with a hole in it can be a water clock, and while the bucket’s interior can be said to be in a modeling relationship with time, the bucket is not a world model. It is a model of something very local. Training language models on Othello game sequences can imply that LLMs function as a world model because it showcases their ability to learn and internalize the structured dynamics and rules of a complex system, rather than merely memorizing patterns.
> > >
> > > I was not just looking for a definition of "world model" in a response to me on OpenReview, but strongly feel that it should be in the paper. If your enter paper revolves around investigating a "world model hypothesis", it makes no sense to me not to have a 100% clear and unambiguous definition of what that actually means, right in the paper.
> > >
> > > I also feel that it could still be much more precise. "X is a global theory of the world". That's not precise to me. What does this mean? From the rest of your response here, I understand it as something like: "A world model is a function that, given a state, can tell me what all the legal actions are, and given a state plus a legal action, tell me what the next state and the immediate reward will be." That would be a precise definition (don't know if it would be correct?).
> > >
> > > Of course, a **crucial follow-up question** will then be: do your experiments actually test for the definition. I doubt that they do, at least not for the definition I have come up with.
> > >
> > > > We apologize for any confusion regarding the objective our models are trained to achieve. To clarify, the models are trained to predict the next move in a sequence, given the preceding moves. For evaluation, we measure the proportion of predicted moves that are legal within the context of the game. This approach follows the problem setting established in previous work [1]. For example, given the sequence of previous moves ‘D6C6C5,’ the model is expected to predict a move like ‘C4’ for evaluation. We have revised the caption for Table 1 and improved the description in Section 3 to ensure clarity and avoid further misunderstandings.
> > >
> > > I also still don't think this is precise enough. A game (like Othello) is not just a sequence of moves. It's a set of rules by which we can play, and any individual play is a sequence of such moves, leading to an outcome as defined by the rules. Given a sequence of moves, just saying that you predict a single next move is an ill-defined problem. There could be many different next moves. If you say that you do this **for a single specific player** (maybe even an optimal one), or a set of players, sure, that works. This needs to be 100% clear from the text though. And I have strong doubts (given my understanding of "world model") that this tests for world models.

---

> ### Author Response · Authors · 2024-11-22
> **Note on world models**
>
> A world model is a representation or a map of a world, i.e., ideally, a homomorphism. We could have been more explicit about this, but this is the standard interpretation of 'world model' in the LLM understanding debate. This should also be clear from the fact that we say our (ideal) world model is the Othello board layout: The world model we evaluate for is a map of an Othello board. In lines 291-2, we say: 'To validate the Othello World Model Hypothesis, we directly evaluate the internal representation of
> the Othello board in language models.' For the evaluation of world models, we check the cosine distance under a Procrustes analysis (see §4.2). Since a homomorphism is invariant under linear projection, this directly evaluates whether our candidate world model is indeed a map of the world (the Othello board).
>
> **We've added a corresponding section in Appendix A, in case readers are not familiar with this area.**

---

> ### Author Response · Authors · 2024-11-23
> **Response by Authors**
>
> > If it's not shown in the figure, I can't see it, and I also can't agree with any conclusions that are derived from it. If you do have data on this, please just include it in the figure.
>
> Sure. We've improved the figure and add the results in the figure.
>
> > In the figure, it is evident that the performance of non-pretrained models, such as GPT-2 and Flan-T5, remains less changed when increasing the data size from 12k to 22k.
>
> We agree with your observation that non-pretrained models achieve strong performance more quickly, whereas pretrained models exhibit slower progress, potentially due to the interference of their pretrained language representations with game understanding. To clarify, we are not suggesting that non-pretrained models perform worse than pretrained models; rather, we are highlighting the differences in their learning curves. We will revise the text to prevent any misunderstanding. Also, we apologize for not attaching the data with longer x-axis. However, we've attached it in our next version and revised the text accordingly.
>
> > While Figure 3 does not explicitly show plateauing for non-pretrained models—indicating instead a slow improvement near the end of the x-axis—we kindly argue that our claims about the comparative trends between pretrained and non-pretrained models remain valid. Specifically, non-pretrained models exhibit sharp, intermediate performance gains on smaller datasets, whereas pretrained models show a more gradual improvement as data size increases.
>
> We would like to clarify that we do not actually mean to claim that pretrained models are better than non-pretrained one. Actually, we see that pretrained knowledge from upstream natural language tasks poses a negative impact on the othello game understanding (as stated in Line 237). What we mean here is just try to compare the curves between the two settings. However, we updated the corresponding parts in our newest version to avoid misunderstanding.
>
> > We apologize for our lack of precision. This should be the rules of the game - or similar. This was not yet updated in the PDF.
>
> We've improved the corresponding section in our newest version. Thanks for the reminder.
>
> > I also still don't think this is precise enough. A game (like Othello) is not just a sequence of moves. It's a set of rules by which we can play, and any individual play is a sequence of such moves, leading to an outcome as defined by the rules. Given a sequence of moves, just saying that you predict a single next move is an ill-defined problem. There could be many different next moves. If you say that you do this for a single specific player (maybe even an optimal one), or a set of players, sure, that works. This needs to be 100% clear from the text though. And I have strong doubts (given my understanding of "world model") that this tests for world models.
>
> We agree that understanding the game state (specifically, determining which player is currently active) is a critical aspect of game strategy. However, prior research [1,2] has already addressed this problem by training linear and non-linear probes to predict game states using trained models. **Their results demonstrate that a linear projection can achieve near-perfect accuracy in deriving the board state**[2], which they argue supports the world model theory. In our work, we did not conduct experiments specifically targeting this perspective. Instead, we focused on single-move prediction, as previous studies have already provided strong evidence for the learnability of game states. Our goal was to build upon these findings and explore how well models can predict the optimal next move, which is a complementary yet distinct challenge requiring both an understanding of the game state and strategic reasoning.
>
> [1] Li et al. Emergent World Representations: Exploring a Sequence Model Trained on a Synthetic Task.
>
> [2] Nanda et al. Emergent Linear Representations in World Models of Self-Supervised Sequence Models.
>
> **We thank the reviewer again for stating the concerns. We've updated our paper according to the reviewer's suggestions and look forward to further discussion.**

---

> > ### Comment · Reviewer_ezgd · 2024-11-25
> >
> > Unfortunately, I continue to be worried about the clarity of writing in this paper, and the lack of precision.
> >
> > > We apologize for any confusion regarding the objective our models are trained to achieve. To clarify, the models are trained to predict the next move in a sequence, given the preceding moves. For evaluation, we measure the proportion of predicted moves that are legal within the context of the game. This approach follows the problem setting established in previous work [1]. For example, given the sequence of previous moves ‘D6C6C5,’ the model is expected to predict a move like ‘C4’ for evaluation. We have revised the caption for Table 1 and improved the description in Section 3 to ensure clarity and avoid further misunderstandings.
> >
> > I think I understand what you are doing, based on a combination of this response with the paper. I'm still actually not 100% sure though, which is concerning, as this should not be a complicated story at all. It should be possible to make this very precise, clear, unambiguous, and at the same time easy to read, written in plain language. Based on my current understanding, it could go something like: *Given a prefix of actions from the training data, we train the model to predict the next action in the sequence. For testing, we ask the model to generate sequences of actions. Any action that is legal in the corresponding game state is counted as a correct action, and any illegal action is counted as a mistake. In other words, we do not require the model to reproduce sequences of actions from training data, or to produce strong or optimal sequences of actions, but simply sequences of legal actions.*
> >
> > The above is my current understanding of what you are doing. Please correct if wrong, but, regardless of whether I understood correctly or not, please think about how you can  describe what you are doing much more precisely, in plain language.
> >
> > If my understanding is correct, I do now get confused about some of your discussion around the 2-hop move prediction, and your responses to Reviewer yh19 though. You talk in a bunch of places about how this task is relevant due to the strategic reasoning involved. The end of section 3.2 talks about "deeper decision-making process required for gameplay strategies". The same at the end of 3.3. The Limitations section (6) discusses how the 2-hop move prediction would be challenging due to the complexity involved in optimal play, strategies, ..., and it suggests that predicting that far into the future is inherently an underdetermined task as there may be multiple optimal moves due to symmetries. Does this mean that you now actually do care about specifically predicting *good* actions, not just any legal action? This seems to conflict with my understanding of what you are doing as I described above.

---

> ### Author Response · Authors · 2024-11-26
> **Response from the Author**
>
> Thank you for stating your concern again.
>
> > Given a prefix of actions from the training data, we train the model to predict the next action in the sequence. For testing, we ask the model to generate sequences of actions. Any action that is legal in the corresponding game state is counted as a correct action, and any illegal action is counted as a mistake. In other words, we do not require the model to reproduce sequences of actions from training data, or to produce strong or optimal sequences of actions, but simply sequences of legal actions.
>
> Yes, your understanding is correct. The only thing is, in the paper, we use the term 'move' instead of 'action'. **We have revised the corresponding task description sections to make it clearer.**
>
> > If my understanding is correct, I do now get confused about some of your discussion around the 2-hop move prediction, and your responses to Reviewer yh19 though. You talk in a bunch of places about how this task is relevant due to the strategic reasoning involved. The end of section 3.2 talks about "deeper decision-making process required for gameplay strategies". The same at the end of 3.3. The Limitations section (6) discusses how the 2-hop move prediction would be challenging due to the complexity involved in optimal play, strategies, ..., and it suggests that predicting that far into the future is inherently an underdetermined task as there may be multiple optimal moves due to symmetries. Does this mean that you now actually do care about specifically predicting good actions, not just any legal action? This seems to conflict with my understanding of what you are doing as I described above.
>
> We need to clarify that the initial intuition behind testing the model's 2-hop performance is to quickly evaluate whether it can learn "good" strategies beyond merely generating legal moves. Previous studies, as well as our experimental results in Table 1, indicate that the model achieves near "perfect" performance in legal move prediction when trained on large datasets. However, this raises the critical question: does the model truly understand how to play the game? And what level of understanding should we expect from from the model? To address these questions, we assess the model's ability to generate two moves consecutively (also sequences of more than two moves in our preliminary experiments). But we find it's more challenging for the model to generate more than one step, and also, the model has close to zero accuracy to generate the whole legal game sequence. So this triggers our discussion of the limitation of what LLMs really learn in Line 483.

---

### Official Review · Reviewer_yh19 · 2024-11-05

**Soundness:** 3
**Presentation:** 3
**Contribution:** 2
**Rating:** 5
**Confidence:** 4

**Summary:**

In this paper, authors evaluate the Othello World Model hypothesis using different types of language models. This study is based on the previous works Li et al. (2023) and Nanda et al. (2023). The goal of this study is to reevaluate the hypothesis over multiple language models and see common representations they learnt.

**Strengths:**

This work is based on previous studies on Othello World Model Hypothesis. It's an interesting study because it tries to see if language models can model the rules of the Othello game from a large amount of transcripts data. Although the hypothesis has been probed in the previous studies, authors propose to reevaluate the hypothesis with more language models and different settings.

From the reevaluation, authors provide more evidence on the hypothesis and try to provide cross-language model latent representation on the Othello World model.

As a result, the paper could support the previous work's claims with new evidence.

**Weaknesses:**

The weak point of this study is to see the contribution claimed by authors as an important new contribution or extension of the previous work's claims. Although the authors tried to use multiple language models to see the difference of the modeling capability, it's not a new problem formulation because it's based on the previous works.

It's unclear why two-hop move generation is introduced as a new benchmark problem. Authors need to explain how two-hop generation provides insights beyond one-hop prediction, or to discuss potential limitations of the one-hop approach.

**Questions:**

* Why is the two-hop move generation an important benchmark in the Othello World Modeling?
* Could you please provide detailed analysis on the difference of each language model on the Othello world modeling? Why do they show different behaviors on the task?
* Please discuss potential implications for particular fields or research areas that might benefit from insights into how language models learn structured world representations.
* What is the reason to revisit this hypothesis using more language models and comprehensive probings? Is the previous work not enough to show the hypothesis's validity?
*Please discuss specific types of problems or domains where your approach might be applicable, and what challenges you anticipate in extending beyond Othello.
* Could you show that the Othello world model encodes the rules of Othello (to determine the validity of moves) or strategy of game playing?

---

> ### Author Response · Authors · 2024-11-22
> **Official Comment by Authors**
>
> **Response to weakness**
> > 1. The weak point of this study is to see the contribution claimed by authors as an important new contribution or extension of the previous work's claims. Although the authors tried to use multiple language models to see the difference of the modeling capability, it's not a new problem formulation because it's based on the previous works.
>
>  With all due respect, we disagree strongly with the idea that something isn’t new because it’s based on previous work. Scientific work is always based on previous work (we stand on the shoulders of giants); yet, we made it very clear what our main contribution is (lines 49-85): We present a new evaluation method for the Othello problem; we also evaluate more models, present novel visualizations, and discuss important data limitations. The new evaluation method - quantifying the isometry between the representations of the model and the physical layout - is crucial for establishing the existence of world models, as argued for in the paper.
>
> > 2. It's unclear why two-hop move generation is introduced as a new benchmark problem. Authors need to explain how two-hop generation provides insights beyond one-hop prediction, or to discuss potential limitations of the one-hop approach.
>
> We extended the one-hop step generation setting in the original Othello paper and investigated two-hop move generation for investigating the model’s capability to anticipate more strategic, long-term planning in Othello. While one-hop prediction focuses on the immediate next move based on the current board state, it inherently overlooks the deeper decision-making process required for gameplay strategies. Since Othello is a dynamic game where optimal play often sacrifices short-term gains for long-term advantages, the two-hop generation pushes models to simulate this high-level reasoning and provides insights on how much LLMs really understand the game strategy in the zero-shot way. *Specifically, we’ve made more clarifications in Section 3.2, 3.3 and the Limitation section in our newest version explaining the insights we expect to gain beyond one-hop prediction.*
>
> **Response to Questions**
> > 1.Why is the two-hop move generation an important benchmark in the Othello World Modeling?
>
> Please see my response to weakness 2.
>
> > 2. Could you please provide detailed analysis on the difference of each language model on the Othello world modeling? Why do they show different behaviors on the task?
>
> We’ve provided some case study by showing an example move from Mistral and T5 in Figure 4. We apologize for not listing other models for page limit. **However, we add the predictions by other LLMs for the same game in Appendix.** We find that most of the time, Mistral shows good performance. It consistently demonstrates the best performance across different scenarios, effectively generating legal moves and showing a nuanced understanding of game rules. The Bart model frequently predicts adjacent tiles, leading to numerous failure cases, particularly when trained with smaller datasets. Llama-2 exhibits inconsistent performance, with a tendency to favor certain tile positions or exhibit a bias in move selection. While its predictions are often reasonable, the model appears to lack the robust policy understanding seen in Mistral, especially under constrained training conditions.
>
> > 3. Please discuss potential implications for particular fields or research areas that might benefit from insights into how language models learn structured world representations.
>
> The induction of world models enables out-of-domain inferences and bias estimation. In addition, investigating the parallels between how language models learn structured representations and how humans internalize similar concepts can shed light on the cognitive processes underlying reasoning, strategy, and language. This could deepen our understanding of human cognition and inform theories of learning and representation. *We've added a corresponding section in our newest version to discuss the impacts.*

---

> > ### Author Response · Authors · 2024-11-22
> > **Official Comment by Authors (1)**
> >
> > > 4. What is the reason to revisit this hypothesis using more language models and comprehensive probings? Is the previous work not enough to show the hypothesis's validity? *Please discuss specific types of problems or domains where your approach might be applicable, and what challenges you anticipate in extending beyond Othello.
> >
> > As stated in the paper, this is exactly our motivation: Previous work was not enough to show that Othello training induces a board model. As it's limited in one small-scaled model, GPT-2. This leaves several important questions. For instance, it remains unclear whether their findings generalize to larger-scale language models or how much training data is required to achieve "perfect" performance. Additionally, their study does not explore whether differences in model architecture could yield similar levels of game understanding. More broadly, we extend this line of inquiry by probing whether language models understand the game's strategy or merely its rules. To address this, we train models to generate sequences comprising multiple moves at a time, pushing beyond simple rule-based learning. Our experiments reveal that different language models, regardless of their architecture, exhibit high similarity in the learned features. This finding provides additional support for the Othello world model theory, suggesting that language models can internalize representations of game rules and strategies through exposure to simple game sequences. *We've added a future work section and the potential impact section to discuss this.*
> >
> > > 5. Could you show that the Othello world model encodes the rules of Othello (to determine the validity of moves) or strategy of game playing?
> >
> > We demonstrate that the Othello model encodes the rules of Othello—potentially forming a "world model"—through several key findings: 1. **Previous Evidence of Rule Learning**: Prior studies [1,2] have shown that GPT-2 can acquire game rules using accuracy evaluations and linear/non-linear probing methods. Expanding on this, we analyze a broader range of LLMs and find that all models, whether employing an encoder-decoder or decoder-only architecture, achieve strong performance in generating legal moves when trained on extensive game data. 2. **Training-Free Probing with Feature Alignment**: To further explore rule learning, we adopt a training-free probing approach. Using a feature alignment algorithm originally developed for multilingual word alignment, we investigate the similarity of features learned across different models. Results in Table 3 and Figure 4 reveal a consistent pattern of feature distributions, suggesting that diverse models converge on similar representations when trained with Othello game moves. 3. **Latent Move Projections and Physical Position Knowledge**: To assess the extent of rule learning, we analyze latent move projections in Figure 6. These projections show that all legal moves consistently receive high probabilities, while tiles in close physical proximity exhibit high similarity scores. This surprising finding provides robust evidence that LLMs can internalize game policies and develop an understanding of spatial relationships, even without explicit training for this purpose.These results collectively highlight that LLMs are capable of encoding Othello's rules and strategies, offering insights into their potential to form structured representations of complex systems.
> >
> > ----------
> > [1] Li et al. Emergent World Representations: Exploring a Sequence Model Trained on a Synthetic Task.
> >
> > [2] Nanda et al. Emergent Linear Representations in World Models of Self-Supervised Sequence Models.
> >
> > We thank the reviewer again for the suggestions. We've added more elaborations and experimental results concerning the problems discussed in our newest version. We sincerely hope the reviewer can consider these revisions during the rebuttal phase and kindly reassess the overall score.

---

> > ### Author Response · Authors · 2024-12-02
> > **Response to Reviewer yh19**
> >
> > We thank the reviewer once again for their valuable feedback and look forward to further discussion. We would appreciate any additional comments they may have.

---

### Meta-Review · Area_Chair_iEgE · 2024-12-23

**Metareview:**

This paper suffers from somewhat unclear definitions and otherwise lack of details. It is also not super novel. Taken together, this suggests that it is not ready to be published in a selective venue - it probably needs another round of clarifications.

**Additional Comments On Reviewer Discussion:**

The authors did engage productively with the reviewers, but the reviewers were not fully convinced.

---

### Decision · Program_Chairs · 2025-01-22

Reject